Concentration and quantification of Tilapia tilapinevirus from water using a simple iron flocculation coupled with probe-based RT-qPCR

Taengphu Suwimon 1
Kayansamruaj Pattanapon 2
Kawato Yasuhiko 3
Delamare-Deboutteville Jerome 4
http://orcid.org/0000-0002-2574-284X Mohan Chadag Vishnumurthy 4
Dong Ha Thanh 5 htdong@ait.ac.th
Senapin Saengchan 1 6 saengchan@biotec.or.th
1 Fish Health Platform, Center of Excellence for Shrimp Molecular Biology and Biotechnology (Centex Shrimp), Mahidol University , Phayathai, Bangkok , Thailand
2 Center of Excellence in Aquatic Animal Health Management, Faculty of Fisheries, Kasetsart University , Chatuchak, Bangkok , Thailand
3 Pathology Division, Nansei Field Station, Fisheries Technology Institute, Japan Fisheries Research and Education Agency , Minami-Ise, Mie , Japan
4 WorldFish , Bayan Lepas, Penang , Malaysia
5 School of Environment, Resources and Development, Asian Institute of Technology , Klong Luang, Pathum Thani , Thailand
6 National Center for Genetic Engineering and Biotechnology (BIOTEC), National Science and Technology Development Agency (NSTDA) , Klong Luang, Pathum Thani , Thailand
Kistler Whitney
Electronic publication date: 2022 Apr 18
Publication date: 2022
Volume: 10
Electronic Location ID: e13157
Received 2021 Jul 12; Accepted 2022 Mar 2
Copyright: © 2022 Taengphu et al.
Copyright year: 2022
Copyright holder: Taengphu et al.
License: This is an open access article distributed under the terms of the Creative Commons Attribution License, which permits unrestricted use, distribution, reproduction and adaptation in any medium and for any purpose provided that it is properly attributed. For attribution, the original author(s), title, publication source (PeerJ) and either DOI or URL of the article must be cited.
License URL: https://creativecommons.org/licenses/by/4.0/

Keywords: Tilapia lake virus, TiLV, Iron flocculation, RT-qPCR, Quantification, Viral concentration, Environmental RNA, eRNA

Funding: CGIAR Trust Fund This research was conducted under the CGIAR Research Program on Fish Agri-Food Systems (FISH) led by WorldFish and as part of the CGIAR Initiative “Protecting human health through a One Health approach”, which are both supported by contributors to the CGIAR Trust Fund (https://www.cgiar.org/funders/). The funders had no role in study design, data collection and analysis, decision to publish, or preparation of the manuscript.

==============================
Background

Tilapia tilapinevirus, also known as tilapia lake virus (TiLV), is a significant virus that is responsible for the die-off of farmed tilapia across the globe. The detection and quantification of the virus using environmental RNA (eRNA) from pond water samples represents a potentially non-invasive and routine strategy for monitoring pathogens and early disease forecasting in aquaculture systems.

Methods

Here, we report a simple iron flocculation method for concentrating viruses in water, together with a newly-developed hydrolysis probe quantitative RT-qPCR method for the detection and quantification of TiLV.

Results

The RT-qPCR method designed to target a conserved region of the TiLV genome segment 9 has a detection limit of 10 viral copies per µL of template. The method had a 100% analytical specificity and sensitivity for TiLV. The optimized iron flocculation method was able to recover 16.11 ± 3.3% of the virus from water samples spiked with viral cultures. Tilapia and water samples were collected for use in the detection and quantification of TiLV disease during outbreaks in an open-caged river farming system and two earthen fish farms. TiLV was detected from both clinically sick and asymptomatic fish. Most importantly, the virus was successfully detected from water samples collected from different locations in the affected farms (i.e., river water samples from affected cages (8.50 × 103 to 2.79 × 105 copies/L) and fish-rearing water samples, sewage, and reservoir (4.29 × 103 to 3.53 × 104 copies/L)). By contrast, TiLV was not detected in fish or water samples collected from two farms that had previously experienced TiLV outbreaks and from one farm that had never experienced a TiLV outbreak. In summary, this study suggests that the eRNA detection system using iron flocculation, coupled with probe based-RT-qPCR, is feasible for use in the concentration and quantification of TiLV from water. This approach may be useful for the non-invasive monitoring of TiLV in tilapia aquaculture systems and may support evidence-based decisions on biosecurity interventions needed.

Introduction

Tilapia tilapinevirus, commonly known as tilapia lake virus (TiLV), has been recently discovered and is the sole virus in the genus Tilapinevirus and the family Amnoonviridae (International Committee on Taxonomy of Viruses, 2019). TiLV is an RNA virus with a 10 segmented negative sense single stranded genome of approximately 10.323 kb (Bacharach et al., 2016). Since its discovery in 2014, the virus has significantly impacted tilapia aquaculture worldwide (Eyngor et al., 2014; Ferguson et al., 2014; Jansen, Dong & Mohan, 2019). TiLV typically results in a cumulative mortality from 20% to 90% (Behera et al., 2018; Dong et al., 2017a; Eyngor et al., 2014; Ferguson et al., 2014; Surachetpong et al., 2017). There are 16 countries that have confirmed the presence of TiLV, to date (Jansen, Dong & Mohan, 2019; Surachetpong, Roy & Nicholson, 2020). However, it is believed that the disease has a wider geographical spread due to the active movements of live tilapia between countries (Dong et al., 2017b). The waterborne spread of TiLV may contribute to the dissemination of the pathogen to new areas and other fish species (Chiamkunakorn et al., 2019; Eyngor et al., 2014; Jaemwimol et al., 2018; Piamsomboon & Wongtavatchai, 2021). Experimental evidence has already demonstrated that TiLV is transmitted horizontally through water (Eyngor et al., 2014).

Environmental DNA and environmental RNA (eDNA and eRNA) have been demonstrated to be an effective tool in the surveillance of fish pathogens (Haramoto et al., 2007; Kawato et al., 2016; Minamoto et al., 2009; Nishi et al., 2016). Environmental RNA may aid in disease surveillance, as described by Kawato et al. (2016) who used an iron flocculation method to concentrate red sea bream iridovirus (RSIV) in a challenge model with Japanese amberjack (Seriola quinqueradiata). Results from that study showed that detection RSIV by qPCR in fish-rearing water samples peaked more than 5 days before fish mortality occurred, suggesting the potential benefit of the iron flocculation method for forecasting disease outbreaks. Others studies used a cation-coated filter method to detect the DNA of cyprinid herpesvirus 3 (CyHV-3), also known as koi herpesvirus (KHV), from concentrated river water samples 3 to 4 months before mass mortality events occurred in wild carp in Japan (Haramoto et al., 2007; Minamoto et al., 2009). The virus continued to be detectable in river water samples for at least 3 months after the outbreaks (Minamoto et al., 2009). These findings helped local authorities and farmers to make rapid decisions for emergency harvesting, the implementation of biosecurity measures, and appropriate disinfection procedures and fallowing periods.

Several molecular methods have been developed for the detection of TiLV, including RT-PCR (Eyngor et al., 2014), nested and semi-nested PCR (Dong et al., 2017a; Kembou Tsofack et al., 2017; Taengphu et al., 2020), RT-qPCR (Tattiyapong, Sirikanchana & Surachetpong, 2018; Waiyamitra et al., 2018), loop-mediated isothermal amplification (LAMP) (Kampeera et al., 2021; Phusantisampan et al., 2019; Yin et al., 2019) and the nanopore-based PCR amplicon approach (Delamare-Deboutteville et al., 2021). However, all of these methods require fish tissues for diagnosis and none reported any use for TiLV detection from environmental water samples. Previous probe-based RT-qPCR methods have been developed to detect TiLV from tilapia clinical samples with detection limits of 2.7 × 104 or ~70,000 copies (Kembou Tsofack et al., 2017; Waiyamitra et al., 2018). However, these may not be sensitive enough to detect low viral loads of TiLV in environmental water samples. Moreover, at the time of earlier primer and probe design, there were a limited number of TiLV genome sequences in the NCBI database. As a result, sequence variation among viral isolates and within the genome segments may not be accounted for in the design of those previous methods. The objective of this study was to develop a new RT-qPCR assay based on updated publicly available TiLV genomic sequences data, in order to detect and quantify TiLV in fish tissues and in environmental RNA (eRNA) concentrated from fish-rearing water samples using an iron flocculation method.

Materials and Methods

Development of a probe-based quantitative RT-qPCR method for TiLV

Primer & probe design and establishment of PCR conditions

A hydrolysis probe-based RT-qPCR method was developed and optimized for the detection and quantification of TiLV following the MIQE guidelines (Bustin et al., 2009). Segment 9 of the 10 segments of the TiLV genome, was reported to have a relatively high identity (97.44–99.15%) among various TiLV isolates (Pulido et al., 2019). The primers and probe were manually designed based on the conserved regions of TiLV genome segment 9 following multiple sequence alignments of all of the complete coding sequences (n = 25) retrieved from the GenBank database at NCBI as of November 2021 (Fig. S1). Primer TiLV-S9-qF (5′-CTA GAC AAT GTT TTC GAT CCA G-3′) had a 100% match with all retrieved sequences while primer TiLV-S9-qR (5′-TTC TGT GTC AGT AAT CTT GAC AG-3′) and probe (TiLV-S9-qP; 5′-6-FAM-TGC CGC CGC AGC ACA AGC TCC A-BHQ-1-3′) had one mismatched nucleotide from the compared sequences (Fig. S1). The size of the amplified product was anticipated to be 137 bp. The primers and probe were synthesized by Bio Basic Inc. (Canada).

The specificity of the primers and probe was assessed in silico using the Primer-BLAST program (https://www.ncbi.nlm.nih.gov/tools/primer-blast/). Gradient PCR reactions with annealing temperatures ranging from 55 to 65 °C were used to determine the optimal temperature of the designed qPCR primers. The final composition of the optimized TiLV RT-qPCR 20 µL reaction consisted of 1X master mix (qScript XLT 1-Step RT-qPCR ToughMix Low ROX buffer) (Cat no. 95134-500; Quanta Bio, Beverly, MA, USA), 1.5–2 µL (≤300 ng) of RNA template, 450 nM each of forward and reverse primers, and 150 nM of Seg9-TaqMan-Probe. The cycling conditions included a reverse transcription step at 50 °C for 10 min, then an initial denaturation step at 95 °C for 1 min followed by 40 cycles of 95 °C for 10 s and 58 °C for 30 s. RT-qPCR amplification was conducted using the Bio-Rad CFX Connect Real-Time PCR machine.

Construction of a positive control plasmid

A positive control plasmid (pSeg9-351) was constructed in a previous study by Thawornwattana et al. (2021). Briefly, a 351 bp-TiLV segment 9 open reading frame (ORF) product was obtained from an RT-PCR amplification using TiLV-S9-F (5′-ATG TCA CGA TGG ATA GAA-3′) and TiLV-S9-R (5′-TCA TAA AGT TTT ATC GCC AG-3′) primers (Pulido et al., 2019). The RNA extracted from TiLV infected tilapia was used as a template. The amplicon was purified before being cloned into the pGEM T-easy vector (Promega, Madison, WI, USA). The sequence of the recombinant clone was verified using the Sanger technique (Macrogen, South Korea). The obtained pSeg9-351 plasmid was used as positive control and used in RT-qPCR analytical sensitivity assays. The TiLV copy numbers in the stock vials were determined using an online calculator (http://www.scienceprimer.com/copy-number-calculator-for-realtime-pcr) based on the pSeg9-351 concentration (ng) and length (bp), and was then adjusted with sterile water to 106 copies/µL working concentration.

Analytical sensitivity and specificity tests

The analytical sensitivity of the Seg9-targeted RT-qPCR was investigated using 10-fold serial dilutions of pSeg9-351 plasmid template from the 106 to 1 copies/µL template. The assays were performed in duplicate. Standard curves were prepared by plotting the log10 of serial plasmid dilutions vs. quantification cycle (Cq) values. Viral copy numbers in each tested sample were calculated by extrapolating the Cq values to the generated standard curve using the equation and Cq values:

Viral copy number=10(Cq - Intercept)/Slopei.e.,10(Cq–(−42.295))/−3.476

The specificity of the method was tested with RNA extracted (150 ng/reaction) from clinically healthy tilapia, 15 common fish bacterial pathogens, and fish tissues infected with nervous necrosis virus (NNV), infectious spleen and kidney necrosis virus (ISKNV), or scale drop disease virus (SDDV) (Table S1).

Validation of the RT-qPCR assay

We assessed the Seg9 RT-qPCR assay against RNA extracted from 65 samples held in our laboratory. Forty-four samples originated from known TiLV outbreaks and 21 from known non-diseased samples (healthy tilapia). Diagnostic test results were obtained using semi-nested RT-PCR methods as described by Dong et al. (2017a) and Taengphu et al. (2020). The diagnostic specificity and sensitivity of the assay were calculated according to formulas described by Martin (1984): Sensitivity % = [number of true positive samples/(number of true positive samples + number of false negative samples)] × 100

Specificity % = [number of true negative samples/(number of true negative samples + number of false positive samples)] × 100

Optimization for viral concentration protocol

Virus preparation

The viral stock used in this study was isolated from TiLV-infected Nile tilapia using the E-11 cell line, a clone of the cell line SSN-1 derived from the whole fry tissue of snakehead fish (Cat no. 01110916-1VL; Sigma-Aldrich, St. Louis, MO, USA). The virus was propagated as described by Dong et al. (2020). Briefly, 200 µL of TiLV stock (~108 copies/mL) was added into a 75 mL cell culture flask containing a monolayer of E-11 cell and 15 mL of L15 medium (Leibovitz), incubated at 27 °C for 5 days. The culture supernatant containing viral particles was collected after centrifugation at 15,000g for 10 min at 4 °C. The viral stock was kept in aliquots of 1 mL at −80 °C until used.

Iron flocculation

The viral concentration was determined using the iron flocculation method, following the protocol previously described by Kawato et al. (2016) with some modifications. The workflow of this method is illustrated in Fig. 1. Briefly, 100 µL of TiLV viral stock containing ~107–108 viral copies was added to 500 mL of sterile water that contained 1% marine salt and 36 µM ferric chloride. The viral copy numbers were quantified by RT-qPCR using RNA extracted from viral stock vials. The suspension was stirred at room temperature for 1 h before being mechanically filtered through a 0.4-μm pore size polycarbonate filter (Advantec, Chiba, Japan) with a vacuum pump connected to a filter holder KG-47 (Advantec, Chiba, Japan) under <15 psi pressure. The flocculate-trapped filters were then subjected to nucleic acid extraction using the Patho Gene-spin DNA/RNA extraction kit (iNtRON Biotechnology, Gyeonggi-do, South Korea). In comparison studies, the flocculate-trapped filters were soaked in an oxalate-EDTA buffer to re-suspend the trapped particles (John et al., 2011) prior to nucleic acid extraction. Experiments were carried out in two to four replicates. The viral concentration and percentage (%) recovery of the virus copies were calculated from Cq values after flocculation and compared to that of the starting viral stock.

Figure 1 Workflow of TiLV flocculation, concentration and quantification used in this study.

An iron flocculation method was used to concentrate viruses from water (A). The water suspension containing the virus was filtered through a 0.4-μm pore size polycarbonate membrane filter with a vacuum pressure pump (B and C). The flocculate-trapped filter (D) was then resuspended in oxalate-EDTA buffer (E) prior to nucleic acid extraction (F) and TiLV quantification (G).

Tilapia and water samples used in this study

Fish farms

Once the viral concentration method was optimized, we tested the technique from fish and water samples collected from six tilapia farms in 2020–2021. One of the three TiLV outbreak cases occurred in a river’s floating cage from a farm producing hybrid red tilapia, Oreochromis sp. (Farm 1) and two occurred in earthen ponds culturing Nile tilapia, O. niloticus (Farms 2 and 3). Samples were collected during TiLV outbreaks on Farms 1 and 2 and when the disease severity decreased at Farm 3. Three other fish farms had no abnormal mortality reported at the time of sample collection. Farms 4 and 5 had previously experienced TiLV outbreaks, and Farm 6 had never experienced a TiLV outbreak.

Sample collection & preparation

Fish specimens (whole body from fingerlings or internal organs from juvenile and adult fish) were preserved in Trizol reagent (Invitrogen, Carlsbad, CA, USA) and kept on ice during transportation and shipped to our laboratory within 24 h. Water samples of 500 mL per sample per location were collected from fish ponds, kept on ice, and transferred to the laboratory within 24 h. Fish samples, snails, and sludge were collected from the TiLV outbreak case on Farm 2. Water samples from fish ponds, reservoirs, and sewage (outgoing waste water from ponds) were also collected.

Upon arrival at our laboratory, the fish specimens were processed for RNA extraction and the water samples were centrifuged at 5,000 g for 5 min to remove suspended matter before being subjected to iron flocculation and subsequent nucleic acid extraction using the Patho Gen-spin DNA/RNA extraction kit (iNtRON Biotechnology, Gyeonggi-do, South Korea). Viral detection and quantification were then performed to determine the presence of TiLV using the Seg 9 RT-qPCR assay described above. The plasmid template pSeg9-351 was used in a positive control reaction while nuclease-free water was used for the negative control.

Results

A new probe-based RT-qPCR method for detection and quantification of TiLV

The Seg9 RT-qPCR method developed in this study had a sensitivity of 10 copies/µL with mean Cq ± SD values of the detection limit at 38.24 ± 0.09 (Fig. 2A). Therefore, samples with a Cq value ≥38.15 were considered TiLV negative or under the limit of this detection method. The established RT-qPCR was found to be highly efficient with slope = −3.476, R2 = 0.998, and E (amplification efficiency) = 94.0% based on the standard curve analysis (Fig. 2B). The formula, copy number = 10(Cq - Intercept)/Slope i.e., 10(Cq – (−42.295))/−3.476, may be used to calculate TiLV copy numbers found in the assayed samples. The analytical specificity test revealed that the method was highly specific to TiLV alone since no amplifications were found when the method was assayed with RNA templates extracted from three other viruses, 15 bacterial species, and healthy tilapia (Fig. 2C, Table S1). The method had a 100% diagnostic specificity and a 100% diagnostic sensitivity when assayed with previously diagnosed TiLV infected and non-infected fish samples (n = 65 with Cq value ranges 13.02–34.85) (Table 1).

Figure 2 Performance of the newly established probe-based RT-qPCR detection of TiLV genomic segment 9.

(A) Analytical sensitivity assay determined using serial dilutions of plasmid DNA containing a 351-bp TiLV segment 9 insert. Amplification results were from two technical replicate tests. (B) A standard curve was derived from the assays in (A) showing an amplification efficiency (E) of 94.0%. (C) Analytical specificity test of the RT-qPCR protocol against RNAs extracted from common pathogens of fish and healthy-looking tilapia as listed in Table S1. (D) TiLV quantification from template extracted from stock virus (S) and flocculate-trapped filters (F) with resuspension step using two replicates. (E) TiLV quantification from fish samples collected from an outbreak open cage. (F) TiLV quantification from water samples collected from an outbreak open cage. P, positive control; N, no template control; RFU, relative fluorescence units.

Table 1 Diagnostic specificity and sensitivity of the Seg9 probe-based RT-qPCR method.

Test results	Diseased samples (n = 44)	Non-diseased samples (n = 21)	
Positive (+)	True positive 44	False positive 0	
Negative (−)	False negative 0	True negative 21	
Diagnostic sensitivity (%)	100	
Diagnostic specificity (%)	100	

Conditions for viral concentration and percentage recovery

The percentage recovery of TiLV after iron flocculation without suspension of the membrane filter in the oxalate-EDTA buffer was only 2.04 ± 0.5% (n = 2) compared to the original viral stock (Table 2). This was significantly improved with an additional suspension step of the flocculate-trapped filters into oxalate-EDTA buffer prior to RNA extraction. The percentage recovery of TiLV increased to 16.11 ± 3.3% (n = 4) in viral concentration after iron flocculation (Table 2). Figure 2D showed the representative results of viral quantification in water samples using Seg 9 RT-qPCR assays of TiLV after iron flocculation with the resuspension step.

Table 2 Percentage (%) recovery of TiLV from water using iron flocculation method with or without a resuspension step of flocculate-trapped filters soaked in oxalate-EDTA buffer.

Conditions	Number of replications	% Recovery	
Without resuspension step	2	2.04 ± 0.5	
With resuspension step	4	16.11 ± 3.3*	
Note:

* Representative RT-qPCR results are depicted in Fig. 2D.

TiLV detection and quantification from tilapia and water samples

The results of TiLV detection and quantification from tilapia tissues and water samples from different farms and water sources are shown in Tables 3 and 4. In the first disease outbreak (Farm 1; river open-cages), TiLV was detected from both fish and water samples from all four cages (A–D) (Table 3). Fish samples had Cq values ranging from 12.40 to 36.22, which was equivalent to 3.98 × 108 to 5.6 × 101 viral copies/150 ng RNA template, respectively (Table 3, Fig. 2E). Eight water samples collected from four cages in close proximity in the same water body had Cq values ranging from 31.19 to 36.76, equivalent to a viral load ranging from 3.40 × 105 to 8.50 × 103 viral copies/L, respectively (Table 3, Fig. 2F).

Table 3 Quantification of TiLV from fish and water samples during an active outbreak in river open-cages.

Cage in Farm 1	Samples	Cq	TiLV load*	Interpretation	
A	Diseased fish A1-1 (liver + spleen)	13.02	2.64 × 108	+	
Diseased fish A1-2 (liver + spleen)	30.69	2.18 × 103	+	
Diseased fish A1-3 (liver + spleen)	13.11	2.49 × 108	+	
Water sample A1	36.76	8.50 × 103	+	
Water sample A2	31.95	2.06 × 105	+	
B	Diseased fish B1-1 (liver + spleen)	14.35	1.10 × 108	+	
Diseased fish B1-2 (liver + spleen)	17.49	1.37 × 107	+	
Diseased fish B1-3 (liver + spleen)	13.13	2.46 × 108	+	
Water sample B1	32.54	1.39 × 105	+	
Water sample B2	31.60	2.59 × 105	+	
C	Diseased fish C1-1 (liver + spleen)	14.76	8.34 × 107	+	
Diseased fish C1-2 (liver + spleen)	13.87	1.50 × 108	+	
Water sample C1	32.71	1.24 × 105	+	
Water sample C2	31.49	2.79 × 105	+	
D	Diseased fish D1-1 (liver + spleen)	36.22	5.6 × 101	+	
Diseased fish D1-2 (liver + spleen)	12.40	3.98 × 108	+	
Diseased fish D1-3 (liver + spleen)	18.67	6.26 × 106	+	
Water sample D1	35.90	1.50 × 105	+	
Water sample D2	31.19	3.40 × 105	+	
Note:

Gray highlights water samples; An asterisk (*) indicates the viral copy (per reaction for 150 ng fish extracted RNA & per L of water sample); A plus (+) indicates the virus detected.

Table 4 Quantification of TiLV from fish and pond water samples from earthen ponds.

Farm	Pond	Samples	Cq	TiLV load*	Interpretation	
Farm/Hatchery 2 (Active TiLV outbreak)	Fingerling pond C1 (TiLV affected pond)	Fish	Diseased F1 (liver + spleen)	12.42	3.93 × 108	+	
Diseased F2 (liver + spleen)	14.56	9.53 × 107	+	
Diseased F3 (liver + spleen)	12.11	4.83 × 108	+	
Diseased F4 (liver + spleen)	10.77	1.17 × 109	+	
Diseased F5 (liver)	13.46	4.17 × 108	+	
Healthy looking F1 (whole fish)	29.85	3.80 × 103	+	
Water	Location 1	39.73	–	–	
Location 2	33.30	8.41 × 104	+	
Snail	Pooled sample	−	−	−	
Sludge	Pooled sample 1	−	−	−	
Pooled sample 2	−	−	−	
Fingerling pond C2 (No signs of TiLV)	Fish	Healthy looking F1 (whole fish)	−	−	−	
Heathy looking F2 (whole fish)	32.88	5.11 × 102	+	
Water	Location 1	34.66	3.42 × 104	+	
Location 2	39.76	−	−	
Fingerling pond C3 (No signs of TiLV)	Fish	Healthy looking F1 (whole fish)	37.34	2.6 × 101		
Healthy looking F2 (whole fish)	−	−	−	
Water	Location 1	−	−	–	
Location 2	−	−	–	
Broodstock pond B1 (No signs of TiLV)	Fish	Female brood 1, Healthy looking#	37.08	3.10 × 101		
Female brood 2, Healthy looking#	35.42	9.50 × 101		
Male brood 1, Healthy looking#	38.28	−	−	
Male brood 2, Healthy looking#	36.18	5.70 × 101		
Water	Location 1	37.79	4.29 × 103	+	
Broodstock pond B2 (No signs of TiLV)	Water	Location 1	−	−	−	
Location 2	−	−	−	
Broodstock pond B3 (No signs of TiLV)	Water	Location 1	−	−	−	
Location 2	−	−	−	
Sewage	Water	Location 1	34.61	3.53 × 104	+	
Location 2	−	−	−	
Reservoir	Water	Location 1	−	−	−	
Location 2	37.78	4.32 × 103	+	
Farm 3 (Active TiLV outbreak)	Fish	Survivor F1 (spleen)	36.45	4.80 × 101	+	
	Survivor F2 (spleen)	37.22	2.88 × 101	+	
	Water	Location 1	35.08	2.59× 104	+	
	Location 2	39.03	−	−	
	Location 3	35.90	1.50× 104	+	
Farm 4 (With history of TiLV outbreak)	Fish	Healthy F1 (whole fish)	−	−	−	
	Healthy F2 (whole fish)	−	−	−	
	Water	Location 1	39.27	−	−	
	Location 2	−	−	−	
Farm 5 (With history of TiLV outbreak)	Water	Location 1	39.18	−	−	
	Location 2	38.24	−	−	
	Location 3	−	−	−	
Farm 6 (No history of TiLV outbreak)	Water	Location 1	−	−	−	
	Location 2	−	−	−	
Note:

Gray highlights water samples; An asterisk (*) indicates the viral copy (per reaction for 150 ng fish extracted RNA & per L of water sample); #, liver, kidney, spleen, gill, gonad; A negative (−) indicates that the virus was not detected; A positive (+) indicates that the virus was detected.

In the second disease event (Table 4, Farm 2), samples were collected from eight ponds; one had unusually mortality (C1), five showed no sign of disease (C2–C3, B1–B3), one was a sewage pond and one a reservoir pond. In the affected fingerling pond C1, TiLV was detected from five diseased fish (9.53 × 107 to 1.17 × 109 copies/150 ng RNA template), one asymptomatic fish (3.80 × 103 copies/150 ng RNA template), and water sample from one location (8.41 × 104 copies/L) (Table 4, Farm 2).

TiLV was undetectable from snail and sludge samples originating from pond C1. TiLV was detectable in the seven other remaining ponds of Farm 2 in relatively low viral loads from some asymptomatic fish (both fingerling and brood fish). TiLV was also detectable in water from culture ponds C2 and B1 and water from the reservoir and sewage ponds that were collected during the disease event (Table 4, Farm 2). In case of TiLV outbreak on Farm 3, both survivor tilapia were positive for TiLV (Cq 36.45–37.22), and two out of the three water samples contained TiLV at 1.50 × 104 to 2.59 × 104 viral copies/L (Table 4). TiLV was not detected in samples taken from these farms or from Farm 6 with no history of TiLV infection despite the fact that Farms 4 and 5 had experienced a TiLV outbreak a few years earlier (Table 4).

Discussion

Methods to concentrate and recover viral particles from environmental water samples have been applied in human health studies especially with waterborne diseases caused by enteric viruses (Cashdollar & Wymer, 2013; Haramoto et al., 2018). The process has now become essential in the study of aquatic environments (Jacquet et al., 2010). Several techniques have been used for viral concentration from aquatic environment, including coagulation/flocculation, filtration/ultrafiltration, and centrifugation/ultracentrifugation (Cashdollar & Wymer, 2013; Ikner, Gerba & Bright, 2012). Our study used an iron flocculation method which was initially described for virus removal from freshwaters (Chang et al., 1958) and virus concentration from marine waters (John et al., 2011). It was later adapted to detect and quantify two fish viruses: nervous necrosis virus (NNV) (an RNA virus) and red sea bream iridovirus (RSIV) (a DNA virus) that were experimentally spiked in fish-rearing water (Kawato et al., 2016; Nishi et al., 2016). The recovery rate was estimated by qPCR and yielded >50 and >80% for NNV and RSIV, respectively. In this study, the recovery rate of TiLV (an RNA virus) from spiked-water was considerably lower (16.11 ± 3.3%), however, similar practical methods have been used for concentrating and detecting human viruses from water environments (Haramoto et al., 2018). For example, murine norovirus-1 (MNV-1) used as a viral model in viral concentration assay of human enteric viruses was recovered from spiked-water at 5.8–21.9% using the electronegative hydroxyapatite (HA)-filtration combined with polyethylene glycol (PEG) concentration method. This protocol was then used for the detection of human noroviruses (NoV) and hepatitis A virus (HAV) in all water types (De Keuckelaere et al., 2013). More recently, researchers used porcine coronavirus (porcine epidemic diarrhea virus, PEDV) and mengovirus (MgV) as model viruses to concentrate severe acute respiratory syndrome coronavirus 2 (SARS-CoV-2) from water samples (Randazzo et al., 2020). The use of an aluminum hydroxide adsorption-precipitation concentration method, allowed PEDV and MgV spiked in water to be recovered at 3.3–11.0%. The method can then be applied to detect SARS-CoV-2 RNA in untreated wastewater samples of ~105.4 genomic copies/L (Randazzo et al., 2020).

Despite a low recovery rate from water samples in this study, we confirmed the usefulness of the iron flocculation and RT-qPCR approach to concentrate and determine the concentration of TiLV from fish-rearing water and other water sources from two aquaculture production systems during disease outbreaks. The inherent nature of DNA and RNA viruses and their ability to persist outside their hosts may also contribute to the differences observed in the recovery rates (Cashdollar & Wymer, 2013; Pinon & Vialette, 2018). Other viral concentration techniques using different coagulant/flocculant chemicals, as well as more efficient RNA extraction methods, should be tested for further improvement of TiLV recovery from water.

Following the viral concentration and recovery processes, viral detection is generally performed using PCR-based assays, cell culture methods, or viral metagenomics analysis (Haramoto et al. (2018)). Here, we employed the RT-qPCR technique for the detection and quantification of TiLV, although the detected amounts did not represent the viral viability. Using all TiLV genomic sequences publicly available, we designed a new set of conserved primers and a probe targeting the viral genomic segment 9. The newly established RT-qPCR protocol was highly specific to TiLV and did not cross-amplify RNA extracted from other common bacterial and viral aquatic pathogens. The method is very sensitive as it can detect as low as 10 viral copies per µL of template, which is >2,700 times more sensitive than previous probe-based RT-qPCR methods (Kembou Tsofack et al., 2017; Waiyamitra et al., 2018). Our RT-qPCR method has a 100% diagnostic specificity and sensitivity in agreement with previous results (n = 65) obtained using semi-nested RT-PCR protocols (Dong et al., 2017a; Taengphu et al., 2020). An increased number of sample sizes with diverse geographical sources may be required for further investigation. Most importantly, this new Seg 9 RT-qPCR assay was able to detect and quantify the TiLV load from various types of field samples, including clinically sick fish, asymptomatic fish, and water samples, as opposed to other molecular diagnostic methods optimized solely for use in fish specimens.

The viral loads from water samples collected during the two disease events were approximately ~103 viral copies/L from earthen ponds and ~104 viral copies/L from open-cages. However, these concentrations may be significantly higher due to substantial losses during the concentration and recovery process. Higher viral loads observed in some of the water samples collected during the disease outbreak may be due to the active shedding of the virus from diseased fish into the environment, and may confirm the waterborne transmission nature of TiLV that was reported previously (Eyngor et al., 2014; Yamkasem et al., 2019). The potential to forecast TiLV outbreaks should be further investigated by experimental infection to monitor viral loads in water in relation to fish morbidity and mortality as previously described for other fish pathogens (Haramoto et al., 2007; Kawato et al., 2016; Minamoto et al., 2009; Nishi et al., 2016).

Conclusions

In summary, the viral concentration method by iron flocculation used in concert with a newly developed probe-based RT-qPCR was not only successful for the detection and quantification of TiLV from water in diseased pond/cages, but also from unaffected ponds, reservoir, and sewage water. This method, apart from its potential practical use for future monitoring programs of TiLV viral load in water samples from various culturing units, may be useful to detect possible TiLV contamination from incoming and outgoing waste water as well as to test the systems after disinfection treatments. Such applications will support health professionals and farmers to design appropriate biosecurity interventions to reduce the loss caused by TiLV in tilapia farms and hatcheries.

Supplemental Information

Supplemental Information 1 Sample used for evaluation of analytical specificity and sensitivity of the probe-based RT-qPCR method.

Click here for additional data file.

Supplemental Information 2 Nucleotide sequence alignments of TiLV segment 9 sequences (n = 25) retrieved from the GenBank database at NCBI.

Accession numbers and viral isolate names of all 25 sequences are shown on the left panel. Position of primers and probe used in the newly developed RT-qPCR assay are marked. Numbers denote nucleotide positions to the putative coding region.

Click here for additional data file.

Supplemental Information 3 Raw data for diagnostic specificity and sensitivity of the Seg9 probe-based RT-qPCR method (Table 1) and percentage recovery of viruses from water using different conditions (Table 2).

Click here for additional data file.

The authors would like to thank K. Pimsannil, W. Meemetta, and T. T. Mai for their skilled technical assistance.

Additional Information and Declarations

Competing Interests

Author Contributions

Animal Ethics

Data Availability

The authors declare that they have no competing interests.

Suwimon Taengphu performed the experiments, analyzed the data, prepared figures and/or tables, authored or reviewed drafts of the paper, and approved the final draft.

Pattanapon Kayansamruaj performed the experiments, authored or reviewed drafts of the paper, and approved the final draft.

Yasuhiko Kawato conceived and designed the experiments, authored or reviewed drafts of the paper, and approved the final draft.

Jerome Delamare-Deboutteville conceived and designed the experiments, authored or reviewed drafts of the paper, and approved the final draft.

Chadag Vishnumurthy Mohan conceived and designed the experiments, authored or reviewed drafts of the paper, and approved the final draft.

Ha Thanh Dong conceived and designed the experiments, analyzed the data, prepared figures and/or tables, authored or reviewed drafts of the paper, and approved the final draft.

Saengchan Senapin conceived and designed the experiments, performed the experiments, analyzed the data, prepared figures and/or tables, authored or reviewed drafts of the paper, and approved the final draft.

The following information was supplied relating to ethical approvals (i.e., approving body and any reference numbers):

No ethical approval was required because the study dealt with fish carcasses and was not involved with live animal operations.

The following information was supplied regarding data availability:

The data are available in the Supplemental Files.

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
