# Peer review of "Concentration and quantification of Tilapia tilapinevirus from water using a simple iron flocculation coupled with probe-based RT-qPCR"

_PeerJ, doi:10.7717/peerj.13157_

## Round 0.1 · original submission · Major Revisions

This study is a well-conducted study; however, it is missing one major requirement for publication. The authors report the performance of a new RT-qPCR assay, but they did not follow the MIQE guidelines are required by PeerJ. There are other minor issues that have been noted throughout the annotated PDF. There are also several comments made by the reviewers that should be addressed.

Reviewer 1 ·

Basic reporting

The purpose of this study is to detect tilapia tilapinevirus (TiLV) in water collected from fish farming environments. Tilapia tilapinevirus is a novel RNA virus that affects tilapia production worldwide. Thus, the study is helpful in managing tilapia farms before and during the virus outbreak. The testing of TiLV in water collected from tilapia ponds or cages suggested that TiLV could be detected in the water.

Experimental design

- Although the objective of the study is to detect the virus in water, nonetheless, the main part of the manuscript focus on the development of a new probe-based real-time PCR method. Previously, numerous real-time PCR and RT-PCR protocols have been developed to detect this virus. Any justification why the authors decide to develop a new assay? If the reason is the matter of sensitivity of the previously developed assay, however, to come to that conclusion, different PCR methods should be tested with the same samples or laboratory.
- In this study, samples were collected from two fish farms. Additional samples taken from different farms/outbreaks will help confirm whether the viral load in the water reflects the amount of virus in the fish.

Validity of the findings

- The inconsistency of virus detection in sick fish, healthy fish, and water collected from the specific ponds raises the concern of the accuracy of the data. More testing of water samples (both from the outbreak and no outbreak situation) is likely to increase the reliability of the findings, which should support the conclusion that TiLV detection in water could be used as an indicator to forecast the outbreak status. In detail, the level of TiLV in disease fish ranges from undetectable to low and high detectable levels (Cq as low as 10). However, the viral loads in the water collected from the close/open cages were in the same range (31 to negative), no matter the viral load in the fish. One should expect that a high viral load should be detected in the pond water during the outbreak. In contrast, low detectable or absence of TiLV is present in water with apparently normal fish.

Additional comments

- Delete extra “,” in line 150

·

Basic reporting

The article is very useful for TiLV disease management and biosecurity. The article is very clear writing in all parts with professional English used. The methodologies used were in a professional standard level. I have only few general comments to the authors as follow :

Line 113 ; What is actual number of sequences used for multiple allignment ? 25 or 27 or Both ?
Line 269-271 ; Please write it more clear for this sentence.

Experimental design

The research question was well defined and meaningful.
The investigation have been conducted with high technical standard.
The methodologies used were professional and standard.

Validity of the findings

The article is very useful, impact and novelty for TiLV disease management and biosecurity.

Additional comments

I have encourage the authors to further improve the viral concentration method or find new methods for concentration of virus in the future. It will be more impacted and very useful to fish farm biosecurity in the case that they have very low viral concentration in rearing water.

·

Basic reporting

Ln 3.1 TiLV is an RNA virus. The notion that it can be detected from eDNA is misleading
Ln 113 ’25 or 27’ the context needs more explanation as to how it was applied
Ln 118 use of respectively not clear
General note: need for sentence construction refining

Experimental design

Ln 134 authors need to describe in more detail the preparation of the standard curve and the plasmid that was used
Ln 137 this section should have been used as a validation step for the developed assay as concordance testing. Why did authors use a subheading that could mean their assay when not?
Ln 160 the authors have not mentioned how virus titration was done but still go ahead to give the viral titer (107 - 108) how exactly was this done?
Ln 164-165 the reason for subjecting to different treatments should be explained at this stage
Ln 186 the manufacturer of the kit used should be indicated
Ln 194 Cq of 38.24 is not very reliable!
Ln 170-183 the description seems not to be in line with the main study indicated in the title and the objective. Why are the fish samples being included? The objective should be adjusted to include this or this component should be removed. In addition, the relationship between the submitted pond/cage water and assay conducted is not clear. Was it used immediately after submission? Was the assay developed earlier and then applied to screen the water samples? Was it kept prior to analysis? What were the conditions? How long should a water sample be kept before analysis? What storage conditions should be used?
These questions should be answered by the authors to guide the prospective users of this screening technique
Ln 205 all the standard deviations used in this manuscript are so wide! Authors need to crosscheck their analysis.

Validity of the findings

no comment

Additional comments

General note: authors are mixing clinical disease investigation and with assay development. It’s advisable that they concentrate on clear description of assay development without mixing it with clinical investigation (diseases fish tissues). Ln 213-230 shows clear deviation from the title and objective set out by the authors.

Compliment: I have been struggling to raise TCID50 of 107 by cell culture in E11 cells. Its impressive how the authors could get 108. It would be nice to include the quantitation method that was used in these assays and how the procedure was conducted to guide readership.
Table 2: not clear, crowded and hard to understand
Fig 1 legend does not correspond with subtitles in the figure. Read ‘a’ in the figure and ‘a; in the legend. Communicating different information.
Fig2: as observed previously, the process leading to the generation of the standard curve should be described in detail.

---

## Round 0.2 · Minor Revisions

The authors addressed the majority of the concerns from the previous version. However, the writing is difficult to read. I have gone through and edited large sections of the manuscript. Read through and address the changes.

---

## Round 0.3 · Minor Revisions

The science of the article is not a problem. However, the manuscript needs to be edited for language. It has improved from the last version, but still needs more work.

---

## Round 0.4 · accepted · Accept

Excellent work having the manuscript edited.